# Landauer Bound and Continuous Phase Transitions

**DOI:** 10.3390/e25070984

**Published:** 2023-06-28

**Authors:** Maria Cristina Diamantini

**Affiliations:** NiPS Laboratory, INFN and Dipartimento di Fisica e Geologia, University of Perugia, Via A. Pascoli, I-06100 Perugia, Italy; cristina.diamantini@pg.infn.it

**Keywords:** Landauer bound, continuous phase transitions, analog computing

## Abstract

In this review, we establish a relation between information erasure and continuous phase transitions. The order parameter, which characterizes these transitions, measures the order of the systems. It varies between 0, when the system is completely disordered, and 1, when the system is completely ordered. This ordering process can be seen as information erasure by resetting a certain number of bits to a standard value. The thermodynamic entropy in the partially ordered phase is given by the information-theoretic expression for the generalized Landauer bound in terms of error probability. We will demonstrate this for the Hopfield neural network model of associative memory, where the Landauer bound sets a lower limit for the work associated with ‘remembering’ rather than ‘forgetting’. Using the relation between the Landauer bound and continuous phase transition, we will be able to extend the bound to analog computing systems. In the case of the erasure of an analog variable, the entropy production per degree of freedom is given by the logarithm of the configurational volume measured in units of its minimal quantum.

## 1. Introduction

Landauer’s principle [1,2,3] tells us that forgetting is costly: the erasing of one bit of information, namely resetting it to a particular memory state, independently of its previous memory state, has an entropic cost of, at least, kTln(2) energy (where T is the temperature and k the Boltzmann constant). This is the content of the famous statement that “information is physical” as realized first by Szilard [4] and after by Landauer: information can only be processed by physical systems, computers, and thus it is subject to the laws of thermodynamics of physical systems. The minimum energy expenditure of kTln(2) solves the problem of the violation of the second law of Maxwell’s demon [5]: the second law is not violated since one has to take into account the cost of erasing the demon’s memory. The paradox of Maxwell’s demon has also been addressed in a related but slightly different way by Brillouin [6,7], using the idea of negentropy, namely the reverse of entropy, describing a system becoming “more ordered”, and its relation with information. A bit of information is obtained by the demon at the price of some negative entropy lost in the environment, which allows the demon to make choices which decrease the entropy in the environment. The relation between the negentropy and the Landauer limit was analyzed in [8,9] considering a system of magnetic skyrmions. There, it was shown that the Landauer bound can be seen as a variation of the negentropy of the skyrmion. Landauer’s principle was recently experimentally verified in [10,11,12,13].

Since its formulation, many discussions have been devoted to the validity and usefulness of Landauer’s principle [14,15,16], and many attempts have been devoted to possibly beat Landauer’s limit since it sets a minimum energy expenditure in computation. More sophisticated formulations have been proposed [17,18], which take into account the role of the conditional entropy to relate Shannon and Gibbs entropy and that lower Landauer’s limit. For a review on recent developments on the thermodynamics of information, see [19]. Another possibility to beat this limit is to admit errors during the erasure procedure. In this case, the original Landauer limit and the associated minimum cost of erasure can be lowered. In [20], it was shown that, admitting errors, the Landauer bound can be lowered and the minimum work necessary to stochastically erase one bit becomes
(1)$$\frac{-\Delta S}{k}=\ln2+p\;\ln\left(p\right)+\left(1-p\right)\;\ln\left(1-p\right)\;,$$
where *p* is the error probability. This error probability, which can be interpreted as mutual or conditional entropy [21,22], becomes relevant [20] for future nanoscale implementations of switches which must necessarily take into account also their thermal fluctuations.

Information is physical; this statement, as shown in [23], implies that physical systems, which contain order, can encode information bits. Continuous phase transitions represent a paradigmatic example of these physical systems. Continuous phase transitions are characterized by symmetry breaking. The order parameter describes the symmetry broken phase and it is zero in the unbroken phase. These transitions, which are generally driven by temperature in the classical case, are described by the phenomenological Landau theory expressed in terms of the temperature and of the order parameter [24]. As we lower the temperature below the critical temperature $T_{C}$, at which the phase transition takes place, the Landau function [24], representing the effective potential, goes from one single minimum to a manifold of minima, e.g., it bifurcates into two minima in the case of $Z_{2}$ symmetry breaking, creating a new order and a new configuration space for the system. The work performed on the system to lower the temperature is completely used to lower the entropy and to change the state of the system, making it “more ordered”. We call this procedure efficient. To better understand the relation between Landauer erasure and continuous phase transition, let us consider one classical bit of information stored in a bistable potential, exemplified by a particle in a double potential well. The first step of the erasure of the memory corresponds to the lowering of the barrier; note that in this step, the phase space available doubles. This corresponds to the disordered phase with m=0. Then, in general, by applying a tilting force and raising again the barrier will force the particle to be in one of the wells, depending on the tilting force, thus resetting either to zero or to one. This is a non-equilibrium state valid for a time smaller than the relaxation time in the well. In this last step, we have a phase space reduction; we ‘compress’ two states in one, and this is what causes the heat dissipation. In spontaneous symmetry breaking, something similar happens. Above the critical temperature, all possible degenerate ground states are available, while below the critical temperature, the system ‘chooses’ one state. In the case of symmetry breaking, an external perturbation, which is generally then set to zero, is what makes the system choose a particular state. In Landauer erasure, this is the role of the tilting in the double-well potential model for a single switch [10].

A particularly interesting example is given by neural networks [25], which are composed of a large number of interacting stochastic bits. Neural networks are the basic elements of associative memories. Contrary to address-oriented memories, recovery of the information is based on the similarity between the stored memory pattern and the presented pattern. The Hopfield model [26] is the most used example of neural networks [25]. The Hopfield model undergoes a phase transition, characterized by order parameter *m* that goes from zero in the disordered phase to one in the ordered phase. The transition is driven by a fictitious temperature: below $T_{C}$, the system becomes ordered. As shown in [23] (and derived in Section 2) this phase transition is akin to an erasure with errors for the N stochastic neurons. The entropy difference between the disordered phase and the partially ordered phase can be exactly written as Equation (1) of [10,20,21], where, in this case, the error probability is related to the order parameter. Note that, while in the Landauer case, the erasure corresponds to forgetting, in the case of the neural networks, it corresponds to remembering.

A phase transition is clearly a collective phenomenon; a single spin cannot have a phase transition, while one can erase a single bit or flip a single spin. However, in the example we chose of the Hopfield model, which, as we will show in the next section, can be mapped in a long range Ising model [25], the bits which compose the associative memory represent the information bearing degrees of freedom. The order parameter *m*, which characterizes the continuous phase transition, plays the role of the error in an erasure with errors. When the parameter *m* is equal to one, we have the completely ordered phase corresponding to the erasure without errors. In [23], we showed that the entropy difference between the disordered phase and the completely ordered phase goes approximately kTln2 times the number of spins of the network, which coincides with the number of the information bearing degrees of freedom.

The Landauer principle [1] was originally formulated to compute the minimal energy required to erase a bit of information and it applies, thus, to the system in which information is represented by discrete units. What happens for analog computing systems? In [27], the relation between erasure and continuous phase transition allowed us to extend Landauer’s principle to systems where information is a continuous variable.

When we erase discrete information, assuming that the conditional entropy is zero [18], the difference in the Shannon entropy between the final state, to which we reset the memory, e.g., to one, and the one in which the system can be in any one of the possible states $s_{i}$ with probability $p_{i}$, is given by
(2)$$\Delta S_{S}=\sum_{i}^{M}p_{i}\ln p_{i}\;,$$
where *M* is the finite number of possible logic states. The continuous generalization of the Shannon entropy is defined as [28,29]
(3)$$S_{S}^{cont}=-\int_{x\in M}p\left(x\right)\ln p\left(x\right)\;.$$
where p(x) is the probability distribution of the relevant degree(s) of freedom. The information-theoretic continuous Shannon entropy, however, requires an appropriate regularization, which adapts the dimensional character of the relevant degrees of freedom to the dimensionless quantity considered in the probability density p(x). This is because the continuous extension of the Shannon entropy, contrary to the discrete entropy, which is an absolute quantity, is not invariant under the change of coordinates [28]. To cure this problem, Jaynes [30,31,32] proposed to modify Equation (3) by introducing an invariant factor $p_{0}\left(M\right)$, which represents the density of the discrete distribution, which gives p(x) in the continuum limit:(4)$$S_{S}^{cont}=-\int_{x\in M}p\left(x\right)\ln\frac{p(x)}{p_{0}\left(M\right)}\;.$$

The factor $p_{0}\left(M\right)$, introduced as a regularization, arises naturally when we consider the continuous Landauer reset. This factor needs to be introduced to cure the problem of classical continuous entropy, which can be negative and divergent [33,34], and is given by the minimum quantum of configuration volume of the physical system.

In Section 2 of this review, we analyze the relation between continuous phase transitions, characterized by an order parameter, and the Landauer bound [23]. Using the example of the Hopfield model [26], we show that the information-theoretic expression for the entropy production during the erasure process, expressed in terms of the error probability, has the same expressions as the thermodynamic entropy in the partially ordered phase. For the Hopfield model, however, the completely ordered state corresponds to perfect remembering rather than forgetting, so the Landauer bound sets a lower limit for the cost of ‘remembering’ [35].

In Section 3, using the relation between the Landauer’s limit and continuos phase transitions, we extend the results of Section 2 to analog computing systems [27]. In this case, the entropy production per degree of freedom during the erasure of an analog variable is given by the logarithm of the configurational volume measured in units of its minimal quantum. Additionally, in this case, we have a “discretization” of the information bearing degrees of freedom, and an infinite amount of energies will be required to perform a computation with infinite precision.

## 2. Thermodynamic Entropy in Continuous Phase Transitions and Landauer Bound

Neural networks, using the definition given in [25], “are algorithms for cognitive tasks, such learning and optimization, which are in a loose sense based on concepts derived from research into the nature of the brain”. One important task that neural networks perform is pattern recognition: the retrieval of information, contrary to address-oriented memories, is performed by looking at the “similarity” between a pattern, which is presented, and the stored patterns. Associative memories have the advantage of being able to retrieve information even in the case of incomplete or noisy inputs, which is not permitted in traditional computers. The Hopfield model [26,36] is the paradigmatic example of a neural network designed to perform the task of associative pattern retrieval and is largely used in associative memory.

In associative memories, when a new pattern is presented, the network evolves from a totally unknown state to a state which corresponds to the stored pattern. As shown in [25], this is gauged equivalent to a state with all neurons, e.g., equal to +1. The transition between the unknown state and the final state corresponding to the stored pattern is, by definition, the process of remembering rather than forgetting, and the Landauer limit corresponds to the minimum energy necessary for remembering. The noise affects the remembering process: when it is not too large, the network provides the minimum energy required to remember, and when errors become too important, there is a phase transition to a state in which remembering becomes impossible.

The Hopfield model [26] is a directed graph of *N* binary neurons $s_{i}$, i=1…N, with $s_{i}=\pm 1$ fully connected by symmetric synapses with coupling strengths $w_{ij}=w_{ji}$ ($w_{ii}=0$), which can be excitatory (>0) or inhibitory (<0). The state $s_{i}=+1$ indicates the firing state of the neuron, while $s_{i}=-1$ indicates the resting state. The network is characterized by an energy function
(5)$$E=-\frac{J}{2}\sum_{i\neq j}w_{ij}\;s_{i}s_{j}\;,\;\;s_{i}=\pm 1\;,\;\;i,j=1\ldots N\;,$$
where *J* represents the (positive) coupling constant. The dynamical evolution of the network state is defined by the random sequential updating (in time *t*) of the neurons according to the rule
(6)$$\begin{array}{ccc} s_{i}\left(t+1\right) & & =\mathrm{sign}\left(h_{i}\left(t\right)\right)\;, \end{array}$$
(7)$$\begin{array}{ccc} h_{i}\left(t\right) & & =J\sum_{i\neq j}w_{ij}s_{j}\left(t\right)\;, \end{array}$$
where $h_{i}$ is the local magnetization. As is standard for neural networks and, thus, for the Hopfield model [25], the temporal evolution proceeds in finite steps, which correspond to the updating of neurons according to the rule Equation (7) in this model. At time (*t* + 1), the neurons are firing or resting depending on the activation function. This process is intrinsically discrete in time. The synaptic coupling strengths are chosen according to the Hebb rule [25]
(8)$$w_{ij}=\frac{1}{N}\sum_{\mu =1\ldots p}\sigma_{i}^{\mu }\sigma_{j}^{\mu }\;,$$
where $\sigma_{i}^{\mu }$, μ=1…p are *p* binary patterns to be memorized. The synaptic strengths contain all the information of the memory, which is encoded in the interaction between the spins $\sigma_{i}^{\mu }$.

The dynamical evolution of the networks will allow the system, prepared in an initial state $s_{i}^{0}$ (presented pattern), to retrieve the stored pattern $\sigma_{i}^{\lambda }$, which most closely “resembles” the presented pattern, namely the one that minimizes the Hamming distance, i.e., the total number of different bits in the two patterns.

Updating the Hopfield network according to the Hebb rule guarantees that the dynamical evolution minimizes the energy of Equation (5): the stored patterns are “attractors” for this dynamic, namely, they are local minima of the energy functional, which is bounded below. This implies that, when an initial pattern is presented, it will evolve until it overlaps with the closest stored pattern and then not evolve anymore. The possibility of remembering depends, however, crucially upon the loading factor α=p/N, given by the ratio between the number of stored memories and the number of available bits [25]: above a critical value, the network has a phase transition into a spin glass [36], and remembering becomes impossible.

In what follows, we consider the case of a single stored pattern $\sigma_{i}$. As shown in [25], using the gauge transformation,
(9)$$s_{i}\to \sigma_{i}s_{i}\;,$$
the energy functional Equation (5) becomes
(10)$$E=-\frac{J}{2N}\sum_{i\neq j}\;s_{i}s_{j}\;,$$
the Hopfield model thus reduces to the long-range Ising model, and the stored pattern becomes $\sigma_{i}=+1$ for all *i*. Remembering for the network, in this case, is equivalent to resetting the *N*-bit register to this value. Note that in the Hopfield model, since the synapses are quadratic in the spins, there is always a symmetry between the memory and its NOT for one stored pattern, e.g., $s_{i}=1\to s_{i}=-1\;\forall i$, if the stored pattern all spins up as we chose in the present case. Both are minima for the dynamic. However, when a pattern is presented, the system recovers the one that is closed in the Hamming distance to the stored pattern.

The deterministic update law Equation (6) can be made probabilistic, introducing a fictitious temperature T=1/kβ and, thus, thermal noise:(11)$$\mathrm{Prob}\left[s_{i}\left(t+1\right)=+1\right]=f\left[h_{i}\left(t\right)\right]\;,$$
where the activation function *f* is the Fermi function
(12)$$f\left(h\right)=\frac{1}{1+\exp(-2\beta h)}\;.$$

The deterministic behavior is recovered in the limit β→∞. The main difference with respect to deterministic neurons, which are always active or dormant according to the sign of *h* is that stochastic neuron activities fluctuate due to thermal noise and we can define a mean activity for a single neuron:(13)$$\langle s_{i}\rangle =\left(+1\right)f\left(h_{i}\right)+\left(-1\right)f\left(-h_{i}\right)\;,$$
where 〈…〉 denotes the thermal average. Now we note that, for the long-range Ising model, the mean field approximation $\langle f\left(h_{i}\right)\rangle \to f\left(\langle h_{i}\rangle \right)$ is exact [37], and we thus obtain the deterministic equation:(14)$$\langle s_{i}\rangle =\tanh\;\left(\frac{\beta J}{N}\sum_{j\neq i}\langle s_{j}\rangle \right)\;.$$

Defining the mean magnetization as $m\equiv \left(1/N\right)\sum_{i}<s_{i}>$, we can rewrite Equation (14) as
(15)m=tanh(βJm),
where we considered the thermodynamic limit N→∞. We can now apply the known results for the mean field Ising model. The self-consistency Equation (15) has only one solution for βJ<1, which corresponds to zero magnetization, m=0. When βJ>1, Equation (15) admits three solutions m=0 and $m=\pm m_{0}\left(\beta \right)$, but only the second two solutions are stable against small fluctuations; we thus have a magnetization m≠0. The condition βJ=1 gives the critical temperature $T_{c}=J/k$: for $T>T_{c}$, the network is disordered and remembering is not possible, while for $T<T_{c}$, the network exhibits a partial magnetization m≠0, which goes m→1 for T→0. Partial erasure is, thus, possible.

Remembering for stochastic neurons is equivalent to a reset operation with errors. For $T\geq T_{c}$, individual neurons fluctuate freely, and we are in the disordered phase. When *T* goes below $T_{c}$, neurons become partially frozen in the stored pattern configuration and m(T) will tell us what is the average rate of errors in the reset process at this temperature. In this procedure, all work performed by lowering the temperature goes into lowering the entropy of the system. In fact, as *T* goes infinitesimally below $T_{C}$, the Landau function [24] bifurcates into two minima, creating a new order and a new configuration space for the network. Note, however, that if the erasure processes is performed in a finite amount of time, in this case, the system will dissipate a finite amount of heat [38,39].

Following the standard treatment for the mean field Ising model (which in the present case is an exact solution), we expand the spin variables $s_{i}$ around their mean value *m* as $s_{i}=m+\delta s_{i}$, with $\delta s_{i}\equiv \left(s_{i}-m\right)$. At the lowest order, the energy functional becomes
(16)$$E=\frac{JNm^{2}}{2}-Jm\sum_{i}s_{i}\;,$$
where we omitted an irrelevant constant. At this order in $\delta s_{i}$, the partition function is
(17)$$Z=\sum_{\mathrm{conf}.}e^{-\beta E}=e^{-\beta JNm^{2}/2}{\left[2\cosh(\beta Jm)\right]}^{N}\;.$$

We thus obtain for the entropy the expression
(18)$$\begin{array}{ccc} S & & =\frac{\partial }{\partial T}\left(kT\ln Z\right)\\ & & =kN\left[\ln(2\;\cosh(\beta mJ))-\beta mJ\;\tanh(\beta mJ)\right]\;. \end{array}$$

At $T=T_{c}$, m=0, the system is disordered, and the entropy takes the maximum value S=kNln2, while at T=0, m=1 and S=0, the system is ordered and the remembering is perfect. The entropy variation between the disordered state and the state with partial remembering 0<m(T)<1 is
(19)$$\begin{array}{ccc} \frac{-\Delta S}{kN} & & =\frac{1}{kN}\left(S_{T_{c}}-S_{T}\right)\\ & & =\ln2-\ln\left(2\;\cosh\left(\frac{mT_{c}}{T}\right)\right)+\frac{mT_{c}}{T}\;\tanh\left(\frac{mT_{c}}{T}\right)\;. \end{array}$$

Equation (19) represents the heat dissipated per bit during the simulated annealing erasure procedure, and, thus, the Landauer bound for stochastic neurons described by the Hopfield model at temperature T. Perfect remembering, T=0 and m=1, gives back the original bound ln(2). Higher temperature corresponds to erasure with errors, in our case, due to thermal fluctuations in the fictitious temperature and, when *T* reaches $T_{C}$ and *m* becomes 0, the system has a phase transition to a disordered state, and remembering is not possible anymore. In the Landauer erasure, this corresponds to resetting to an unknown state, i.e., setting the probability error p=1/2 in Equation (1).

The previous analysis tell us that the error probability *p* in the Landauer erasure is represented by the stochastic updating rules for the Hopfield network Equation (11). According to Equation (11), the probability that a neuron flips due to thermal noise is
(20)$$\mathrm{Prob}\left[s_{i}\left(t+1\right)=-s_{i}\left(t\right)\right]=\frac{\exp\left[-\beta h_{i}\left(t\right)s_{i}\left(t\right)\right]}{2\;\cosh[\beta h_{i}\left(t\right)s_{i}\left(t\right)]}\;,$$
so the probability that it flips from the desired value +1, since we are resetting to a memory register with all bits +1, to the wrong value −1 is
(21)$$p=\mathrm{Prob}\left[+1\to -1\right]\equiv \frac{1}{2}\left(1-m\right)=\frac{\exp\left(-\beta Jm\right)}{2\;\cosh(\beta Jm))}\;.$$

The maximum error probability p=1/2 corresponds to m=0, the maximally disordered state of the network reached at $T=T_{C}$, while p=0 corresponds to the perfect order for the network with order parameter m=1 at T=0. Inserting Equation (21) into Equation (19), we obtain for the entropy difference, and thus for the dissipated heat, exactly the information-theoretic expression Equation (1):(22)$$\frac{-\Delta S}{kN}=\ln2+p\;\ln\left(p\right)+\left(1-p\right)\;\ln\left(1-p\right)\;.$$

When p=0, the Landauer bound is saturated, and the entropy difference between the increasingly disordered state of the model and its perfectly ordered T=0 state reaches the exact value
(23)ΔS=kNln(2).

Once we reach the value m=0 for the order parameter, which describes the broken symmetry phase, we reach the maximum entropy for the network and we cannot keep disordering the system without violating the Landauer bound and, thus, the second law of thermodynamics. The phase transition, which takes place at $T_{C}$, thus corresponds to the saturation of the Landauer limit. The generalized Landauer theorem states, thus, that the sum of the entropy loss per bit and the one-bit error entropy cannot be lower than the bound kln(2), and it is exactly equal to this bound when the procedure is efficient. When this bound is saturated by the error entropy, resetting (remembering here) is no longer possible, and a phase transition occurs.

The Hopfield model has a discrete $Z_{2}$ symmetry corresponding to a spin 1/2. The generalization to higher-order spins with a classical $Z_{\left(2n+1\right)}$ symmetry, with n=1/2,1,3/2… is, however, straightforward. In the case of a $Z_{\left(2n+1\right)}$ symmetry, Equation (23) becomes
(24)$$-\Delta S=\left(S\left(T_{C}\right)-S\left(0\right)\right)=kN\ln\left(2n+1\right)\;.$$

In the more general case of a continuous phase transition of a system of *N* elementary components with *D* degrees of freedom each, which undergoes a continuous phase transition to a partially ordered phase below a critical temperature, we have only *d* degrees of freedom, which survive in the partially ordered phase, while the others are frozen. The phase transition is characterized by a complex vector of ordered parameters whose norm η rises from 0 in the disordered phase to 1 at zero temperature. The ration between the original degrees of freedom *D* and the one in the partially ordered phase *d* can be written as
(25)$$D/d=q^{n}\;,$$
with *n* being an integer larger than one if D/d is a prime power and *q* a prime number. If D/d is not a prime power, we have n=1 and q=D/d. If we take, for example, the simple case q=2, the phase transition can be seen as the formal “resetting” of dN bits to their standard value, with error probability p(T)=(1−η(T))/2. The entropy change during a generic phase transition is, thus, again given by Equation (22) with N→dN. Otherwise, the Landauer bound would be violated in the ordering process. For q=3, we have trits instead of bits, and the generalization to other values of *q* is straightforward.

## 3. Analog Computing Systems

In analog computing systems, information is encoded in a continuous variable. To compute the entropy change during the erasure of information encoded in a continuous variable, we will use the relation between the Landauer principle and entropy change during continuous phase transitions [23]. We will again assume that the erasure is efficient.

We study the 3-dimensional ferromagnetic classic Heisenberg model, which undergoes a phase transition with spontaneous symmetry breaking of O(3)→O(2) [24], which is described by the Hamiltonian
(26)$$H=-\frac{J}{2}\sum_{<i,j>}s_{i}\cdot s_{j}-H\sum_{i}s_{i}\;,$$
where 〈i,j〉, which denotes the sum over nearest neighbors spin, with *i* that goes from 1 to the number *N* of spins and ${\left|s\right|}^{2}=1$. In this case, the spin orientation, that encodes analog information, can take all values on a sphere of unit radius and we have, thus, a continuum of possible values; for the Ising spins, the orientation is binary, up or down. Since the model is ferromagnetic, we have J>0. *H* is a constant external magnetic field in the $\hat{z}$ direction. The ferromagnetic Heisenberg model undergoes a continuous phase transition [24], characterized by an order parameter *m*, the mean magnetization. For $T\geq T_{C}$, the system is disordered, while for $T<T_{C}$, the phase becomes partially ordered and *m* reaches the value 1 at T=0. As in the case of discrete symmetry, lowering the temperature is akin to an erasure process, and *m* plays the role of the error probability in the reset operation [20].

Following what we did in the previous section, we identify the Shannon entropy of the erasure process in the analog computing system with the entropy variation during the transition from from $T=T_{C}$ to T=0. The entropy variation
(27)$$-\frac{\Delta S}{kN}=\frac{\left(S\left(T_{C}\right)-S\left(0\right)\right)}{kN}\;,$$
gives, thus, the Landauer bound for an analog computing system [23].

We use again the mean field approximation; the mean field Hamiltonian for the Heisenberg model is [24]
(28)$$\begin{array}{ccc} H_{\mathrm{mf}} & & =-J\sum_{<i,j>}s_{i}H_{\mathrm{eff}}+\frac{JNm^{2}}{2}\;,\\ H_{\mathrm{eff}} & & =(Jm+H)\;, \end{array}$$
where *m* is the mean magnetization: $m\equiv \left(1/N\right)\sum_{i}\langle s_{i}\rangle$. The effective magnetic field is the sum of the average magnetic field, generated by all other spins, plus the external magnetic field *H*. As usual, we will take the limit H→0, obtaining, thus, the partition function:(29)$$Z=\exp\left[\beta \frac{JN3m^{2}}{2}\right]\int d^{3}s\delta \left(s^{2}-1\right)\exp\left[\beta J\sum_{i=1}^{N}m\cos\theta_{i}\right]\;,$$
where $\theta_{i}$ is the angle between the spin and the $\hat{z}$ direction and $\beta ={\left(kT\right)}^{-1}$. From Equation (29), we derive the free energy
(30)$$\frac{F}{N}=\frac{Jm^{2}3}{2}-\frac{1}{\beta }\ln\left[\frac{4\pi \sinh\beta mJ}{\beta mJ}\right]\;,$$
and from the free energy, the entropy
(31)$$\frac{S}{kN}=\ln\left[\frac{4\pi \sinh\beta mJ}{\beta mJ}\right]-\beta mJL\left(\beta mJ\right)\;,$$
where L(x)=cothx−1/x is the Langevin function.

Let us now consider the limits for $T=T_{C}$ and T=0 of Equation (36). When $T\to T_{C}$, m→0, and the entropy reaches its maximal value, the logarithm of the volume of the configuration space, namely, the area of a sphere of unitary radius:(32)$$\frac{S(T_{C})}{kN}=\ln\left(4\pi \right)\;.$$

When T→0 and m→1, which corresponds to the perfect reset, the entropy becomes negative and divergent, contrary to the third law of thermodynamics:(33)$$\frac{S(T\to 0)}{kN}\to -\infty \;.$$

This problem is common for various classical systems. One textbook example is the classical harmonic oscillator [40], and, in general, the way to cure this problem is to consider the classical system as the limit of its quantum counterpart.

In the mean field approximation, the quantum ferromagnetic Heisenberg model describes a system of quantum, non-interacting spins $s_{i}$ with (2s+1) components in an external magnetic field $H_{\mathrm{eff}}$. The mean field Hamiltonian for the quantum Heisenberg model is [41]:(34)$$H=-H_{\mathrm{eff}}\sum_{i=1}^{N}s_{i}\;.$$

It describes non-interacting (2s+1) components spins in an external magnetic field $H_{\mathrm{eff}}$, which, again, is the sum of the average magnetic field generated by all other spins plus an external magnetic field *H*. The partition function is
(35)$$\;Z={\left[\sum_{n=-s}^{s}\exp\beta H_{\mathrm{eff}}n\right]}^{N}\;,$$
while the entropy, in the limit in which the external magnetic field H→0, is
(36)$$\frac{S}{kN}=\ln\left[\frac{\sinh(1+\frac{1}{2s})\beta mJs}{\sinh\frac{\beta mJs}{2s}}\right]-\beta mJsB_{s}\left(\beta mJ\right)\;,$$
where $B_{s}\left(\beta mJ\right)$ is the Brillouin function defined as $B_{n}\left(x\right)=\frac{2n+1}{2n}\coth\left(\frac{2n+1}{2n}x\right)-\frac{1}{2n}\coth\left(\frac{1}{2n}x\right)$. In the limit T→0, the entropy Equation (36) goes to zero, $\frac{S(T=0)}{kN}=0$, in agreement with the third law of thermodynamics.

The classical limit of the quantum Heisenberg model [42,43,44] is obtained by properly distributing the (2s+1) values of the quantum spin on the classical sphere of area $4\pi r^{2}$ with *r* that is the dimension of an action and is equal to 1 in our case. If we call $s_{\max}$ the highest weight of the representation in the quantum case, we can define the spin density as [42]
(37)$$\Delta \left(2s_{\max}+1\right)\to 4\pi \;\;\mathrm{for}\;\;s_{\max}\to \infty \;,\Delta \to 0\;\;.$$

To ensure the existence of the infinite spin limit [42], we need, however, to rescale the spin as $s\to s/s_{\max}$. The minimum state volume Δ represents the minimum area in the unitary sphere occupied by a spin. This minimum volume is given by the Heisenberg principle:(38)$$\Delta =\frac{1}{2}\hbar^{2}s_{\max}\;;\hbar s_{\max}\to s_{\mathrm{clas}}\;\;\mathrm{for}\;s_{\max}\;\to \infty \;,\hbar \to 0,$$
with $s_{\mathrm{class}}$, which has the dimensions of an action, and $s_{\mathrm{class}}=1$ in our case. Correspondingly, we define the regularized entropy as
(39)$$\begin{array}{cc} \frac{S}{kN}= & \ln\frac{4\pi }{\left(2s_{\max}+1\right)}\left[\frac{\sinh(1+\frac{1}{2s_{\max}})\beta mJ}{\sinh\frac{\beta mJ}{2s_{\max}}}\right]\\ & -\beta mJB_{s_{\max}}\left(\beta mJ\right)\;. \end{array}$$

The classical limit corresponds to $s_{\max}\to \infty$. When $T=T_{C}$, the entropy is, as before,
(40)$$\frac{S(T=T_{C})}{kN}=\ln\left(4\pi \right)\;,$$
while at T=0, we obtain
(41)$$\frac{S(T\to 0)}{kN}=\ln\left(\frac{4\pi }{2s_{\max}+1}\right)=\ln\frac{1}{2}\hbar^{2}s_{\max}=\ln\frac{1}{2}\hbar \;,$$
where we use Equations (37) and (38) (note that in the last term of Equation (41), *ℏ* is divided by a constant that has the dimension of an action and that it is equal to one). This result tells us that if we want to avoid the entropy divergence, we cannot actually send ℏ→0. In fact, the limit ℏ→0 corresponds to a classical distribution concentrated in regions smaller than the minimum area allowed by the Heisenberg principle. Δ is the factor $p_{0}\left(M\right)$ in Equation (4), and its presence is due to the fact that the continuous Shannon entropy must be regularized in order to make it invariant upon a change of coordinates.

Using Equations (40) and (41), we obtain for the entropy variation
(42)$$\begin{array}{ccc} & & -\frac{\Delta S}{kN}=\frac{S\left(T_{C}\right)-S\left(T=0\right)}{kN}=\\ & & =\ln\left(4\pi s_{\mathrm{clas}}^{2}\right)-\ln\frac{1}{2}\hbar^{2}s_{\max}=\ln\frac{8\pi s_{\mathrm{clas}}}{\hbar }\;\;. \end{array}$$

For the Heisenberg model, we have $s_{\mathrm{class}}=1$, and we thus obtain
(43)$$-\frac{\Delta S}{kN}=\frac{S\left(T_{C}\right)-S\left(T=0\right)}{kN}=\ln\frac{8\pi }{\hbar }\;,$$
(note that the quantity inside the logarithm is dimensionless since $s_{\mathrm{class}}=1$ has the dimensions of an action). Entropy variation Equation (43) represents the analog generalization of the Landauer bound: the entropy change during the erasure process performed by resetting a continuous variable, the spin *s*, to a standard value is given by the available configuration volume (the area 4π in this case) measured in units of the minimum quantum of configuration volume Δ. This implies that, both for digital or analog information, physical systems can encode only a finite countable amount of information [28,45,46], and that information can be manipulated only with finite precision: infinite precision, namely the realization of a truly analog computing system, is forbidden by the laws of physics.

The maximum number of possible logic states that we can associate with the Heisenberg model is
(44)$$N_{l}=\mathrm{int}\left(\frac{8\pi }{\hbar }\right),$$
(int(a) is the integer part of number *a*), while for a generic angular momentum *L*, it is
(45)$$N_{l}=\mathrm{int}\left(\frac{8\pi L}{\hbar }\right),$$
to which we can associate a finite number of bits:(46)$$n=\log_{2}\left(N_{l}\right)\;.$$

For the case of a cube of 5×5×5=125 atoms [47], with an angular momentum per atom of the order of L≈ℏ, for which the interactions between the momenta are such that they behave like a single classical momentum, we have $N_{l}=3140$. From Equation (46), we obtain a number n≈11.6 of bits that can be stored. Under the same assumption, in a system of magnetic nano-dots with a 20 nm side, containing approximately 200 million atoms, we can store up to n=27.6 bits. If we want to perform with this system a perfect Landauer reset, the amount of heat to be dissipated is readily provided by Equation (42): Q≥19.11kT, approximately 30 times what we would have for a binary system reset.

We now want to consider the more general case of the symmetry breaking pattern O(n)→O(n−1). Within the mean field approximation, this generalization is straightforward. From Equation (29), substituting $d^{3}s\to d^{n}s$, we have
(47)$$Z=\exp\left[\beta \frac{JN3m^{2}}{2}\right]\int d^{n}s\delta \left(s^{2}-1\right)\exp\left[\beta J\sum_{i=1}^{N}m\cos\theta_{i}\right]\;,$$
and for the entropy
(48)$$\frac{S}{kN}=\ln\left[2^{\frac{n}{2}}\pi^{\frac{n}{2}}\frac{I_{\frac{n}{2}-1}\left(\beta mJ\right)}{{\left(\beta mJ\right)}^{\frac{n}{2}}}\right]-\beta mJ\frac{I_{\frac{n}{2}}\left(\beta mJ\right)}{I_{\frac{n}{2}-1}\left(\beta mJ\right)}\;,$$
with $I_{\nu }\left(z\right)$, the modified Bessel functions of the first kind.

The entropy difference between the perfectly ordered state at T=0 and the completely disordered one, at $T=T_{C}\;,m=0$, which gives the Landauer bound for the erasure of a O(n) spin *s* is
(49)$$-\frac{\Delta S}{kN}=\frac{S\left(T_{C}\right)-S\left(0\right)}{kN}\;,$$
with
(50)$$\frac{S(T_{C})}{kN}=\ln S_{n-1}\;,S_{n-1}=\frac{2\pi^{\frac{n}{2}}}{\Gamma \left(\frac{n}{2}\right)}\;,$$
where $S_{n-1}$ is the area of the (n−1)-sphere of unit radius and Γ(x) is the Euler gamma function. As for the Heisenberg model, the limit T→0 is singular, and the entropy is negative and logarithmically divergent: (S(T→0)/kN)→−∞.

The regularization of the entropy in this case is, however, more complicated since, contrary to the *O*(3) Heisenberg model, the analytical results are not known [48] for the O(n)-symmetric quantum Heisenberg model, not even in the mean field. Additionally, the definition of the classical limit is not clear. Extending to this case the results obtained for the O(3) case, given by Equation (50), we conjecture that for the O(n)-symmetric case, the entropy change during the erasure process will be given by the available configuration volume, the area of the *n*-sphere, measured in units of the minimum quantum of the configuration volume that in this case will be $\propto \hbar^{n-2}$. In the case of the SU(n)-symmetric (restricted to symmetric representations) Heisenberg model, the possibility of having a positive classical entropy was proposed by Lieb and Solovey [49] using a coherent states approach.

## Data Availability

Not applicable.

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
