# Peer review of "Landauer Bound and Continuous Phase Transitions"

_entropy, 2023, doi:10.3390/e25070984_

Round 1

Reviewer 1 Report (Previous Reviewer 2)

The author has made changes in response to my previous concerns.  Although my concerns have not gone away, it is not my paper and so I will not insist on aligning it with my viewpoint.  I appreciate the author’s exposition and can recommend their review for publication in Entropy after some typos are fixed.

Eq. (13) did not render correctly.

I believe there are two missing “f”s and two missing brackets.

Lines 187, 222, and 245:  “entropy production” should be changed to “change of entropy”.  For a thermodynamically efficient protocol, there is no “entropy production” despite the change in system entropy.  See, e.g., [Riechers and Gu, PRA 104, 012214 (2021)] for further discussion of this point.

Line 49 typo:  “is is”

Lines 59 and 103: I assume “patter” should be “pattern”

Line 123:  I assume “ttractors” should be “attractors”

The inline equation between Eqs. (14) and (15) could use the \langle and \rangle brackets for consistent notation.

Line 167:  What is meant by “i.d.”?   Perhaps this should be “i.e.,”?

Just above Eq. (34):  “Heisemberg” should be “Heisenberg” 

Eq. (35):  The notation should be tweaked so that the limits of the summation are not represented by the same variable that is supposedly summed over.  I.e., the summand should contain some variable in place of “s” that is summed over.

The English is fine besides a few typos.

Author Response

I thank the Referee for his/her positive comments and for pointing out some typos that I corrected on the manuscript.

Reviewer 2 Report (Previous Reviewer 1)

The author has addressed my comments. Therefore I recommend publication.

Author Response

I thank the Referee for his/her positive comments.

Reviewer 3 Report (New Reviewer)

The manuscript "Landauer Bound and Continuous Phase Transition" by Marai Cristina Diamantini continues a line of works that concerns the minimal thermodynamic cost of erasing a bit in spin model where the bit is stored in the phase of the system. Relation to the Hopfield model for neural memory is explored.

As far as I can tell, all the calculations in the manuscript are correct. However, if I understand correctly, the conclusions are not. Let me try to explain my concern. Consider for example Eq.(19). This equations correctly assures that erasing a bit is associated with entropy production of ln(2) per spin in this model. However, a single spin is not a single bit! The bit is encoded in the phase of the system, not in every single spin. Therefore, the cost of erasing a bit is actually N*ln(2), which is very far from the Landauer bound!! This is not surprising: indeed a single bit can in principle used to store a bit, and using a macroscopic number of spins to store a single bit is by no mean an efficient way. The same point repeats throughout the manuscript - the Landauer principle sets a bound on the minimal heat that must be dissipated in erasing a bit: this is an absolute limit for all systems regardless of their size, not a limit per spin!

Apart from this issue, there are many typos. I stopped collecting them at some point, but here is a list of some I have noticed:

1. Above Eq.(1), look for "Can be lowered In [9]" (I think a point is missing).

2. Line 49, "is is".

3. Line 53, "which is general then set to zero"

4. Line 80, "Hopefiled model [14], will show" (probably "we" is missing).

5. Line 103, "patter".

6. Line 118, "in in".

7. Line 123, "ttractor".

8.  Line 125, "bounded by below".

9. Eq.(13), left < is missing twice.

10. Line 188, "Second principle" is probably the second law of thermodynamics?

There are quite some few typos that require editing, but the ideas can be understood.

Author Response

Reviewer 4 Report (New Reviewer)

This is a nice paper dealing with the relationship between the Landauer limit and the continuous phase transitions in different fields of physics. The main message conveyed is that the ordering process that occurs after the passage from the disordered phase to the ordered phase in a continuous phase transition can be regarded as an information erasure subjected to the Landauer bond. The author clearly discusses the relation between the thermodynamic entropy in continuous phase transitions and Landauer’s bond and the analog computing systems where the encoding is by means of a continuous variable.

In my opinion, the choice of the journal is appropriate, the planning of the paper is very good and the main arguments are clearly outlined. For these reasons, certainly this article could merit publication in Entropy.

However, in my view, in the present form, it is not so easy to understand, starting from the Abstract, that this is a Review and not an Original paper. In other words, on a first sight, sometimes it is not easy to distinguish between novel results or novel calculations and results and quantitative derivation that belong to previous literature. This is not a criticism but I think for the average reader it could be important to separate between the original results which could be reported also in a Review article and the results discussed from the literature. For example, all the discussion of Sect. 3 starting from Eq.(9) introducing the gauge transformation of the spin variable in the energy function of the Hopfield model reducing it to an Ising model, the introduction of the activation function as a Fermi function, the derivation of the entropy and so on looks also partially original for a reader and needs a clarification.

Said that, I would suggest to address the following:

1. What about negentropy and its relation with Landauer’s bond and continuous phase transitions?

2. In nature there are also first-order phase transitions characterized by sudden jumps of the physical quantities passing from one phase to the other and generally marked by singularities in the free energy. Could this approach and Landauer’s bound be applied also to first-order phase transitions? If yes, a brief discussion could be helpful.

3.  The author discusses analog computing systems where information is encoded in a continuous variable: it is stated that it is used the relation between the second principle and entropy production during continuous phase transitions derived in [11]. I think that more details on this important link would be helpful for a reader.

4.    The 3-dimensional ferromagnetic Heisenberg model is introduced via the Hamiltonian of Eq.(26). It should be underlined that the spin variable is not a scalar but an operator using another symbol indicating an operator, it is a quantum model.

5. It is stated that the ferromagnetic Heisenberg models undergo a continuous phase transition but this occurs only for dimensionality d > 2, please clarify. In this respect on line 196 there is a pending sentence, please check.

6.  A review paper on this important subject should have a larger list of references. I understand that it is impossible to take into account all the literature and the author properly cites key references on the subject but should not ignore other references (recent and not recent) where Landauer’s limit, information entropy and its relation with thermodynamic and statistical mechanics entropy are discussed.

On Landauer’s limit and related phenomena:

    (a) R. Landauer, Information Is Physical. Phys. Today 44, 23–29 (1991).

    (b) R. Landauer, Dissipation and Noise Immunity in Computation and              Communication. Nature 335,  779–784 (1988).

    (c) R. Landauer, The Physical Nature of Information. Phys. Lett. A 217, 188–    193 (1996).

These papers give a further insight into the profound relation between information and physics and complete the original font already cited. The author could briefly discuss them, e.g., in the Introduction.

On Shannon’s information entropy, its relation with thermodynamic entropy and its application to physical systems, there are other contributions by Jaynes for the continuous case and also by Brillouin who has also studied the relationship between information and entropy:

(a) E.T. Jaynes, Information Theory and Statistical Mechanics I, Phys. Rev. 106, 620–630 (1957).

(b) E.T. Jaynes, Information Theory and Statistical Mechanics II, Phys. Rev. 108,171–190 (1957).

(c) L. Brillouin, Maxwell’s Demon Cannot Operate: Information and entropy. I, J. Appl. Phys. 22, 334-337 (1951).

(d) L. Brillouin, Physical Entropy and Information. II, J. Appl. Phys. 22, 338-343 (1951).

There are also more recent papers dealing with the same subject including applications to condensed matter and topological defects:

(a)   T. Sagawa, M. Ueda. Minimal energy cost for thermodynamic information processing: measurement and information erasure. Phys. Rev. Lett. (2009).

(b) R. Dillenschneider, E. Lutz, Memory erasure in small systems, Phys. Rev. Lett. 102, 210601 (2009).

(c) Y. Jun, M. Gavrilov, J. Bechhoefer, High-Precision Test of Landauer's Principle in a Feedback Trap, Phys. Rev. Lett. 113, 190601 (2014).

(d) J.H. Hong, B. Lambson, S. Dhuey, J. Bokor, Experimental test of Landauer's principle in single-bit operations on nanomagnetic memory bits. Science Adv. 2, e1501492 (2016).

(e) D.H. Wolpert, Stochastic thermodynamics of computation, J. Phys. A Math. Theor. 52, 193001 (2019).

(f) R. Zivieri, Magnetic Skyrmions as Information Entropy Carriers, IEEE Trans. Magn. 58, 1500105 (2022).

(g) R. Zivieri, From Thermodynamics to Information: Landauer’s Limit and Negentropy Principle Applied to Magnetic Skyrmions. Front. Phys. (Lausanne) 10, 769904 (2022).

The author could briefly discuss also these papers in the very introductory part to stress the strong relation between the statistical physics and information theory.

Finally:

-

      Line 28: I would suggest to replace the comma with the colon symbol

-       Eq. (10): I agree that the energy is long-range in terms of the critical behavior but the exchange is short-range in terms of interactions, please clarify.

The English grammar should be checked (e.g. “tell” should read “tells”, “represent” should read “represents” and sometimes the opposite, that is the “s” should be removed, e.g. “undergoes” should read “undergo”).

Round 2

Reviewer 3 Report (New Reviewer)

I thank the author for clarifying the point that any spin in the system is actually a memory bit, and that by going through the phase transition all spins become magnetized (and hence all memory bits are erased).

The only thing I would ask the author is to clarify this, namely to explain this point as clearly as possible -- this is a very subtle and misleading point, and at least for someone coming from my community it is not trivial that in a phase transition still the information is encoded in each of the spins and not the in the total magnetization. 

Author Response

I thank the Referee for his/her positive comments. I clarified, as suggested, the point that any spin in the system is a memory bit. I thank the Referee for raising this point.

Reviewer 4 Report (New Reviewer)

I would like to thank the author for having addressed all my comments and questions. In the present form, the manuscript has improved, is finally set as a review article and, therefore, is ready for the publication. I recommend its quick publication.

Author Response

I thank the Referee for his/her positive comments.

This manuscript is a resubmission of an earlier submission. The following is a list of the peer review reports and author responses from that submission.

Round 1

Reviewer 1 Report

This paper studies the link between the Landauer limit and continuous phase transitions. There seem to be some interesting ideas and therefore I am inclined to recommend the manuscript for publication in Entropy, but there are a nubmer of issues that need to be addressed prior to publication.

I am a bit confused about the dynamics that is chosen for the hopfield model. Why is the dynamics in discrete time? And is there any reason to choose the specific jump rates of Eq. 10 and 20? These assumptions seem to be crucial to arrive at the result of Eq. 21, but are not necessary true for realistic physical systems. Also, it is not clear to me why one can keep the magnetization fixed when calculating the entropy, rather than taking the ensemble average over all magnetizations. These assumptions make it hard for me to fully grasp the importance of the central result.

Furthermore, I am confused by the sentence starting on line 133 'Note that this procedure is efficient ...'. It is true that work is done on the system by lowering the temperature of the system, but this should lead to heat production into a thermal bath. Otherwise, this seems like a violation of the first law of thermodynamics. Furthermore, it is worth noting that erasure processes in a finite amount of time generally need to disipate a finite amount of entropy (see e.g., refs. 1-2 below)

Finally, I have some problems with the interpretation of the quantum mechanical erasure. Although this argument works for the spin memories discussed here, I don't think that it would work in spatial systems, where a collapse of the wave function could, in principle, still lead to a diverging amount of information.

Furthermore, I have some smaller issues regarding presentation that need to be addressed prior to publication:

- On page 2: 'second principle' should be 'second law'.

- on line 248 'where. If we want to...' is gramatically incorrect.

- the references are rather messy. For example, in ref. 28, the author writes 'to appear in commun math phys' while this paper has already been published several years ago. More generally the style seems to be inconsistent at several points.

- recent developments on the thermodynamics of information are mostly ignored. I feel like the authors should at least include a reference to the recent review, ref. 3 below.

1 Proesmans, K., Ehrich, J., & Bechhoefer, J. (2020). Finite-time Landauer principle. Physical Review Letters, 125(10), 100602.

2 Van Vu, T., & Saito, K. (2022). Finite-time quantum Landauer principle and quantum coherence. Physical review letters, 128(1), 010602.

3 Parrondo, J. M., Horowitz, J. M., & Sagawa, T. (2015). Thermodynamics of information. Nature physics, 11(2), 131-139.

Reviewer 2 Report

The manuscript “Landauer bound and continuous phase transitions” by Maria Cristina Diamantini considers Landauer’s lower bound for the energy required to reset a bit, and explores similar expressions in the change of entropy as a system is brought below the critical temperature of a continuous phase transition.  The paper discusses Ising, Hopfield, and Heisenberg models, although the analysis is restricted to the mean-field regime.  The author suggests that the similarity of expressions is more than coincidental.

The paper is well written.

I like the set of ideas involved.

However, I am concerned with some of the stated results and interpretations in the paper.  

My main concern is as follows:

(i) Establishing the bistability needed for memory (which is effectively what happens as you move below the critical temperature, or introduce an energetic barrier) is distinct from (ii) resetting to a particular metastable state (which is the main subject of Landauer’s bound).  It makes sense that there will be some similarities in formulae, since each task is explored via differences in entropies.  However, many of these similarities are superficial and, as I will point out, restricted to the mean-field approximation.  The entropy change during the first task scales with the number of sites/particles, while the entropy change during the second task is independent of the number of sites/particles.  Conflating the two concepts would likely create more confusion than clarity within the community.  Hence, I suggest that the author presents a more careful interpretation of the analysis, where the differences between the two tasks is highlighted.  The author’s contributions would be of much greater value if the similarities and differences in the tasks and formulae can be better contextualized and critically examined.

Let me try to make the concern more grounded in the particulars of the manuscript.

Starting with Eq. (9), we see that there is a single desired stored memory for the model, which corresponds to \sigma_i = +1 for all i, as the author states.  However, it should be pointed out here that the dynamics that allow for this stable “memory” also allow for a stable “false memory”, which corresponds to \sigma_i = -1 for all i.  On line 122, the author states that the magnetization m \to 1 as the temperature T \to 0.  But really, we’re only guaranteed that the absolute value |m| \to 1.  

Is the author suggesting that an initial pattern is chosen at the initial time, which is mostly spin up?  Is that why m \to 1 rather than m \to -1?

If so, this would explain why the true memory would be more likely chosen over the false memory.  But it would also introduce new trouble in the analysis.  In particular, the analogous situation in a typical Landauer-erasure scenario would be to start the system primarily in one of the two energy wells.  This is, effectively, resetting the memory before the protocol even begins.

On the other hand, if the initial state is not biased towards the desired memory, then lowering the temperature below the critical temperature will break the symmetry in an arbitrary fashion—with the zero-temperature state either all up or all down, chosen at random.  This would correspond to randomizing the memory, rather than correctly remembering the up over the down state.

The problem with the comparison boils down effectively to this:  Guaranteeing a particular memory state requires non-equilibrium dynamics, either to prepare an initial nonequilibrium state that will preferentially relax to the desired final nonequilibrium state, or to drive a formerly equilibrium state into the desired final nonequilibrium state.  

On line 135, the author states that “this procedure is efficient in the sense that all work done by lowering the temperature goes into lowering the entropy of the system, without any dispersion into a physical thermal bath.   While I agree that the temperature can be changed with vanishing entropy production, and can thus be thermodynamically reversible, it is not quite accurate to say that this is without any dispersion into a physical thermal bath.  Because, indeed, the energy of the system is changing; and by energy conservation, this energy Q should flow to the bath.  The bath then changes entropy, but only by the compensating amount \Delta S = Q/T necessary for a thermodynamically reversible transformation.

The choice to slowly lower temperature of an Ising model is analogous to slowly introducing the energetic barrier in the typical double-well potential-energy landscape of the Landauer paradigm.  In the former case, we get a double well in equilibrium free energy as a function of magnetization.  In the latter case, we get a double well in potential energy as a function of some degree of freedom.  Either example allows for the metastability needed for memory.  But either also \emph{requires some energetic input to break the symmetry in a particular direction of a preferred final well}.

I think the author is close to making this point, but the connection should be made more rigorous while addressing the above concerns.

To summarize my main concern very simply:

Being magnetized is not the same as being magnetized in a particular direction.  Breaking the symmetry in a preferred direction is what Landauer’s bound is all about.

Eq. (21) is particular to the mean-field model.

Other versions of the Ising model won’t exhibit the same Landaueresque form when comparing the system’s entropy above and below the critical temperature.  However, due to symmetry of global spin flip, even more complicated models \emph{will} have a Landaueresque expression if instead one looks at the difference in entropy between the global equilibrium state (with unbroken symmetry) and the local-equilibrium state with broken symmetry, both at the same sub-critical temperature.

Other serious matters:

The editors should be aware that the paper appears to have significant overlap with two of the author’s previous papers, Refs. [11] and [12].  Nevertheless, the author does appropriately reference these previous works, and the current manuscript refers to itself as a review.  The editors can decide whether this is appropriate.

It should be made clear on the first page that to “erase” a bit means resetting it to a particular memory state, independent of its previous memory state.

Eq. (4):

The denominator p_0(M) should appear within the logarithm, as Jaynes proposed, and as stated in Eq. (3) of Ref. [12].

It may also be worth mentioning though that Eq. (3) of Ref. [12] has its own issues: it is always non-positive since it is a negative relative entropy.  To the contrary, entropy should always be non-negative.  Using the denominator p_0(M) both outside and inside the logarithm returns us to familiar grounds.

Around Line 160, the author says

it is impossible to keep disordering continuously the system without violating Landauer’s bound and, thus, the second law of thermodynamics. The only way out is that a phase transition to an entirely new state of matter takes place at T = Tc.

While enticing, this statement seems misled.

The mean-field model is not physical; if we took it seriously, then indeed there is a conflict in which the probabilities of system configurations would appear to be the same all the way from the critical temperature through infinite temperatures.  This implies that, despite its utility, the mean-field model does not correctly describe entropy at the critical temperature—not even approximately.

For any model that does not have homogeneous all-to-all couplings, the situation is different:  Even above the critical temperature, correlations among sites persist at any finite temperature.  The entropy of the system is thus complicated at and above the critical temperature.  Because of correlation, disorder does not saturate at the critical temperature, and the system is not susceptible to the author’s dilemma in Line 160.

More succinctly:

Entropy saturation does not cause the phase transition as the author suggests in Line 167 but, rather, indicates infinite temperature.

It is true that, at the critical temperature, the entropy of the average spin saturates.  But this ignores all correlations.  Perhaps the author is trying to make the point that some structure below the critical temperature can be understood by average-site behavior, while differences in behavior above the critical temperature must involve differences in correlations among sites.

Again, Eq. (22) is only applicable to the mean-field model.  Accordingly, it seems inappropriate to associate Eq. (22) with Hopfield models.

Around Line 187, the author makes one of their main claims:

The principle, however is always the same: the thermodynamic entropy of a continuous phase transition is a universal expression governed by the Landauer bound.   However, I cannot endorse this interpretation for the reasons I gave above.

Again, the derived expressions only looked like Landauer’s bound because the spins are independent and identically distributed in the mean-field model.  For other models, you won’t get this Landaueresque resemblance for the studied difference S(T=T_c) - S(T=0).

Rather, you’ll only find a general and robust connection with Landauer’s expression when comparing the entropy of the global-equilibrium vs. local-equilibrium states—without and with broken symmetry, respectively—at a given sub-critical temperature. 

How can we make sense of the units within the logarithm in Eq. (38)?  S/k should be unitless, but instead Eq. (38) has the very strange units of log(Energy \times seconds).

This seems to indicate a problem somewhere within the analysis leading up to Eq. (38).

On line 234, the lines between Eqs. (42) and (43), and line 262:

The change in entropy should not be called “entropy production” since these entropy changes can be carried out in a thermodynamically reversible manner, for which entropy production would be zero.

Minor points:

Lines 14 and 16:

Should be “kT ln 2” rather than “k ln 2”.

As a minor notational suggestion, the LaTeX commands “\langle” and “\rangle” would be more standard and aesthetically desirable for denoting averages, rather than “<” and “>”.

Line 192:

It’s not clear what is meant by “the second principle”.

What does the averaged-subscript notation <i,j> mean, in Eqs. (23) and (25)?

Line 246:

“bits, where.” ?